# Migraine patients visiting Chinese medicine hospital: Protocol for a prospective, registry-based, real-world observational cohort study

Shaohua Lyu[1,2☯], Claire Shuiqing Zhang[2☯], Anthony Lin Zhang[2], Jingbo Sun[1], Charlie Changli Xue[1,2]*, Xinfeng Guo[1]*

1 The Second Affiliated Hospital of Guangzhou University of Chinese Medicine, Guangdong Provincial Hospital of Chinese Medicine and Guangdong Provincial Academy of Chinese Medical Sciences, Guangzhou, China, 2 The China-Australia International Research Centre for Chinese Medicine, School of Health and Biomedical Sciences, RMIT University, Bundoora, Victoria, Australia

☯ These authors contributed equally to this work.
* guoxinfeng@gzucm.edu.cn (XG); charlie.xue@rmit.edu.au (CCX)

## Abstract

### Introduction

Migraine is a disabling, recurrent headache disorder with complex comorbidities. Conventional treatments for migraine are unsatisfactory, with side effects and limited effectiveness. Chinese herbal medicine (CHM) has been used as an alternative or complementary treatment option for migraine in China. Currently, the existing evidence of benefit of CHM for migraine has been generated from randomised clinical trials using standardised intervention with a focus on internal validity hence with limited external validity. Moreover, CHM individualised intervention design, patients' preferences and concerns, and clinicians' experience are critical to clinical decision making and therapeutic success. This real-world observational study aims to gather practice-based evidence of effects and safety of CHM for migraine in the context of integrating Chinese medicine diagnostic procedures, patients' preferences and matters relevant to clinical decision making.

### Methods and analysis

The study is being undertaken at the Guangdong Provincial Hospital of Chinese Medicine (GPHCM) from December 2020 to May 2022. We anticipate that approximately 400 adult migraineurs will be enrolled and observed on their migraine severity, analgesic consumption, quality of life, anxiety, depression and insomnia at baseline and then every four weeks over 12 weeks. Treatments, diagnostic information, and patient-reported most bothersome symptoms will be collected from patient clinical records. Patient's demographic data, preferences and concerns on CHM treatments will also be gathered at baseline and be analysed. Factors related to clinical outcomes will be explored with multiple correlation and multivariable regression analyses. Effects of CHM will be evaluated using generalised estimated equation, based on clinical outcome data.

**Funding:** This study was funded by Guangzhou University of Chinese Medicine "Double First-Class" and High-level University Discipline

Collaborative Innovation Team in the form of a grant to JBS (No.2021xk84). The study was also funded by China-Australia International Research Centre for Chinese Medicine (CAIRCCM)-a joint initiative of RMIT University, Australia and the Guangdong Provincial Academy of Chinese Medical Sciences, China in the form of a grant to XFG. The funders had no role in study design, data collection and analysis, decision to publish, or preparation of the manuscript.

**Competing interests:** The authors have declared that no competing interests exist.

**Abbreviations:** AE, Adverse event; CGRP, Calcitonin Gene-Related Peptide; CHM, Chinese herbal medicine; CRF, Case report form; GAD-7, Generalised anxiety disorder 7-item scale; GPHCM, Guangdong Provincial Hospital of Chinese Medicine; ICHD-3, International Classification of Headache Disorders 3rd version; ISI, Insomnia severity index scale; MBS, Most bothersome symptoms; MSQ, Migraine specific quality of life questionnaire; NRS, Numeric rating scale; PHQ-9, Patient health questionnaire-9; RCT, Randomised controlled trial; SNRI, Serotonin Norepinephrine Reuptake Inhibitors; SSRI, Selective Serotonin Reuptake Inhibitors; YLD, Years of life lived with disability.

## Discussion

This study will provide comprehensive evidence of CHM for migraine in the context of evidence-based practice.

## Trial registration number

ChiCTR2000041003

## Introduction

Migraine is a recurrent, severe and primary headache disorder, manifesting unilaterally, throbbing in nature, and lasting 4 to 72 hours [1]. Migraine presented with a global age-standardised prevalence of 14.4% and was ranked as the second most disabling disease in 2016, with an estimated years of life lived with disability (YLDs) of 45.1 million [2]. In China, the number of migraineurs was estimated to be 132 million, with six million YLDs, in 2017 [2, 3].

According to the International Classification of Headache Disorders, 3rd edition (ICHD-3) [1], migraine is typically divided into migraine with aura and migraine without aura. Migraine can also be categorised as episodic or chronic, according to its frequency. However, episodic migraine can progress to chronic migraine at an annual rate of 2.5% [4].

Migraine patients are more likely to suffer from psychiatric disorders including depression and general anxiety disorder [5, 6], as well as sleeping disorders such as insomnia [7]. These comorbidities are more commonly seen in chronic migraine compared with episodic migraine [8], and also in migraine with aura compared with migraine without aura [9, 10]. Furthermore, the comorbidities in episodic migraine indicate a poor prognosis, leading to poorer quality of life and limiting the response to treatments [7, 11], and also increase the risk of progression to chronic migraine and medication-overuse headache [1, 12, 13].

Migraine is usually managed with medications for relieving symptoms in the acute stage, and with prophylactic treatment for reducing the frequency and severity of future attacks. The acute medications include aspirin, nonsteroidal anti-inflammatory drugs, paracetamol, antiemetics, steroids, and triptans. Prophylactic pharmacotherapies include calcium channel blockers, beta-blockers, tricyclic antidepressants, candesartan, pizotifen, angiotensin-converting enzyme inhibitors, selective serotonin reuptake inhibitors (SSRI), serotonin norepinephrine reuptake inhibitors (SNRI), antiepileptics, Botulinum toxin A and calcitonin gene-related peptide (CGRP) [14, 15]. Among them, only tricyclic antidepressants, SSNI and SNRI can be adopted for migraine comorbid with depression [14–16]; Botulinum toxin A is suggested for chronic migraine. Medical devices for migraine include occipital nerve block, vagus nerve stimulation, transcutaneous supraorbital nerve stimulation and transcranial magnetic stimulation [14–16]. However, approximately 30–40% of migraineurs are dissatisfied with current treatments, especially those suffering from chronic migraine [17, 18] and migraine with comorbidities [19–23], citing insufficient treatment effects, low tolerability, and unbearable side effects [24].

Acknowledging the above-mentioned limitations, there is an urgent need to identify alternative or complementary therapies to assist current conventional managements. Chinese herbal medicine (CHM), which is often prescribed to migraine patients in China [25], has gained clinical research evidence as an effective alternative or complementary therapy for migraine in recent years. However, most of the existing evidence has been generated from

randomised controlled trials (RCTs) [26–30] and RCT-based systematic reviews [31–34]. Under controlled settings, RCTs usually apply strict selection criteria and provide unified treatments to all participants [35]. Such design may not reflect the complexity of migraine and its comorbidities [36], nor reflect the real-world practice of CHM which is commonly tailored according to individual's syndrome differentiation [37]. Hence, the research evidence obtained from previous explanatory RCTs and systematic reviews need to be supplemented with more real-world evidence.

To address this research gap, observational studies, which reflect real-world settings and evaluate both standardised outcomes and patient-reported most bothersome symptoms (MBS), can be a pragmatic approach to assess the effects and safety profiles of CHM for migraine and its comorbidities, and provide valuable information for decision making in CHM clinical practice. In line with this, we designed this prospective, registry-based, real-world observational cohort study.

## Objectives

This observational study aims to 1) evaluate the effects and safety of individualised CHM for adult migraine in a real-world setting; 2) identify factors that contribute to clinical outcomes of CHM for migraine; 3) summarise the common patterns of CHM for migraine and its comorbidities; and 4) report on the characteristics of migraine patients who seek CHM management, including their preferences and concerns associated with CHM therapies for their migraine.

## Methods

### Study design

This prospective, registry-based, observational cohort study is being undertaken at the Headache Clinics at the Guangdong Provincial Hospital of Chinese Medicine (GPHCM) in southern China [38]. Participant recruitment commenced in December 2020 and is targeted to end in May 2022. Based on consecutive sampling, an estimated number of 400 eligible migraineurs will be recruited.

All participants will be required to complete case report forms (CRFs) every four weeks over a period of 12 weeks. Following the same period intervals, data regarding participants' detailed treatment plan, including medications, administrative methods, courses, dosage and forms of CHM, which have been determined by their headache specialists, will also be collected from their clinical records. Data will be collected for the entire study duration, even in cases where treatments have been discontinued before the end of the observation period.

The study (version 1.0) was approved by the ethics committee of GPHCM (ZE2020-243-01) and registered with the Human Research Ethics Committee of RMIT University (24235). This study complies with the Declaration of Helsinki, Ethical Guidelines for Medical Research on Humans, and Good Clinical Practice guidelines. The protocol has been registered in the Chinese Clinical Trial Registry (No. ChiCTR2000041003). The reporting of this study will abide by the Strengthening the Reporting of Observational Studies in Epidemiology (STROBE) statement for cohort studies [39].

**Eligibility criteria.** Participants will be eligible for registration if they meet the following criteria:

a). diagnosed with migraine according to the ICHD-3 (1); b). first time seeking treatment for migraine in GPHCM; c). able to provide written informed consent; d). aged of 18 years old or above; and e). not concurrently participating in other interventional trials.

Participants will be excluded for registration and participation if they have any of the following conditions: a). diagnosed with diseases which could induce migraine-like headache such as glaucoma, brain tumour, brain injury; b). diagnosis of migraine being corrected to secondary headaches during the observation period; c). severe impairment of speech, vision, memory or cognition that affects communication and ability to complete questionnaires; d). pregnant or breastfeeding; or e). highly dependent on medical care.

Participants will be marked as drop-outs if they request to withdraw from the observation for any reason.

**Interventions.** Participants will receive individually tailored treatment determined by their headache specialist in the GPHCM; these include CHM in the form of herbal decoction, granules or patented pill, alone or in combination with western medicine or other treatment procedures such as acupuncture. The treatment plan will be based on discussion between individual participants and their headache specialist, with no limitations or interference set by this study.

**Recruitment.** Recruitment posters have been displayed in the waiting room of the Headache Clinics of GPHCM, and headache specialists received training relating to this study have been given screening forms. Potential participants, who have been pre-screened by the headache specialists for eligibility, will be guided to a private consultation room where the researcher or the independent research assistant will provide a full explanation of the study prior to the process of written informed consent. Refusal of participation will not affect the degree of medical care provided.

**Data collection.** The study will employ a structured CRF (S1 File), which has been specifically designed for this study, including demographic information, symptoms, treatment details, relevant validated questionnaires, participants' feedback and adverse events (AEs). Face-to-face data collection will be conducted at baseline, while evaluations at other timepoints can be completed using online CRFs or hard copy CRFs, according to participants' preference. Online data collection will be conducted via the online survey service provider Wenjuan.com (www.wenjuan.com). Participants will be required to take migraine diary and record details of each migraine attack (S2 File). Data of patient-reported MBS, treatments, examinations and diagnostic information will also be extracted from the clinical records by the researcher.

The data to be collected from CRF is outlined below:

*Demographic and general information*. Demographic information including age, gender, general information such as family history, migraine subtypes and other medical histories will be collected upon participant enrolment.

*Patients' preferences and concerns*. Patients' preference and concerns regarding CHM treatment for migraine will be collected upon participants' enrolment.

*Migraine frequency*. Defined as the number of migraine attacks in four weeks. An attack that is interrupted or rescued by sleep or medications but relapses within 48 hours will be considered as one migraine attack; attacks that are separated by an entire 24-hour period will be considered as two distinct attacks [40]. A 50% reduction in migraine frequency within a evaluation interval will be defined as response.

*Migraine days*. Defined as the number of days with migraine headache within each evaluation interval (four weeks) [41].

*Migraine duration*. Defined as the average duration of each migraine attack.

*Pain intensity*. Evaluated via numeric rating scale (NRS), which is an 11-points scale with 0 indicating no pain and 10 indicating the most severe pain. This outcome will provide data regarding the maximal pain intensity prior to taking abortive medication, as well as the average pain intensity across each observational intervals.

*Analgesic consumption.* Data collected under this category include the type, amount and frequency of acute medication, participants' behaviour when purchasing and using analgesics. The quantity of medications will be standardised using medication quantification scales [42], where available.

*Migraine-Specific Quality of Life Questionnaire version 2.1 (MSQ).* A 14-item patient-reporting outcome instrument that measures the impact of migraine on a patient's health-related quality of life. The domains cover role function-restrictive, role function-preventive and emotional function [43].

*Generalised Anxiety Disorder 7-item (GAD-7) Scale.* An efficient, practical, patient-reporting scale, which is commonly used for rapid screening and evaluating the presence and severity of anxiety disorder, especially in outpatient settings [44].

*Patient Health Questionnaire-9 (PHQ-9).* A self-administrative scale to screen, diagnose, monitor and measure the severity of depression [45].

*Insomnia Severity Index (ISI).* A brief self-reporting instrument measuring the patients' perception of both nocturnal and diurnal symptoms of insomnia [46, 47].

*Satisfaction of treatment.* Evaluated using 5-point Likert scale [48] at each observational timepoint.

*AEs.* Participants will be asked to report any AEs possibly related to prescribed treatments, analgesics or concomitant medications, such as diarrhea, dental ulcer, insomnia in CRFs at observational visits.

During data collection, the researcher will report patients' scores of GAD-7, PHQ-9 and relevant AEs to the corresponding clinicians to assist in clinical management.

Migraine diary is a tool for patients to record information of migraine attacks and monitor the frequency, duration, severity, associated symptoms and medication consumption of migraine attacks over time. Data collected from migraine diary will be cross-checked with CRFs as a supplementary method to ensure data accuracy.

Clinical records are referred to collect the following information:

*Treatments, examinations and diagnoses.* Treatment prescriptions and courses, results of laboratory and medical imaging examinations, Chinese medicine syndrome diagnosis, and western medical diagnoses will be obtained from clinical records.

*Patient-reported MBS.* Any other symptoms such as dry mouth, constipation, abdominal distention, which have been reported by participants, will be extracted from clinical records.

Fig 1 provides a summary of data collection during each timepoint of this study. Reminders will be sent via Wechat or via mobile phone messaging system to participants prior to each observational timepoint to improve compliance.

**Estimated number of patients.** This study aims to explore the information related to migraine management based on a registry of the patient cohort from GPHCM available in a specific time range, sample size calculation is not required since there are no pre-determined hypotheses to be tested [49]. This study adopts the consecutive sampling method for recruitment, which is recommended to reduce selection bias in non-randomised studies [49]. The estimated number of participants is approximately 400, based on the average annually outpatient visit numbers at the headache clinic of GPHCM between July 2018 and July 2020.

**Exposures and confounders.** In this real-world observational study, CHM treatment is defined as the main exposure. Participants being treated with CHM solely and those treated with a combination of conventional prophylactic treatment and CHM will be considered as two subcohorts for analyses.

Measured confounders in this study include comorbidities of migraine (depression, anxiety and insomnia, evaluated by PHQ-9, GAD-7, and ISI, respectively), gender, baseline severity of migraine (duration, average frequency, average pain NRS and analgesics consumption at

|  | Baseline | Treatment and Data Collection Timepoints | | |
|---|---|---|---|---|
| **TIMEPOINT** | $T_0$ (Week 0) | $T_1$ (Week 4±1) | $T_2$ (Week 8±1) | $T_3$ (Week 12±1) |
| **ENROLMENT:** | | | | |
| *Eligibility screening* | X | | | |
| *Informed consent* | X | | | |
| *Demographics information* | X | | | |
| *Migraine characteristics* | X | | | |
| *Patients' preferences and concerns* | X | | | |
| *Medical history* | X | | | |
| **ASSESSMENTS:** | | | | |
| *Migraine assessment details* | X | X | X | X |
| *Analgesics consumption* | X | X | X | X |
| *MSQ* | X | X | X | X |
| *GAD-7* | X | X | X | X |
| *PHQ-9* | X | X | X | X |
| *ISI* | X | X | X | X |
| *AEs* | | X | X | X |
| *Satisfaction* | | X | X | X |
| *Treatments and diagnoses* | X | X | X | X |
| *Migraine diary* | X | X | X | X |

**Fig 1. SPIRIT schedule of enrolment and assessments.** AEs: Adverse Events, GAD-7: Generalised Anxiety Disorder 7-item Scale, ISI: Insomnia Severity Index Scale, MSQ: Migraine Specific Quality of Life Questionnaire, PHQ-9: Patient Health Questionnaire-9, $T_0$: Timepoint 0, $T_1$: Timepoint 1, $T_2$: Timepoint 2, $T_3$: Timepoint 3.

entrance), association with the menstrual cycle (pure menstrual migraine vs menstrually related migraine vs non-menstrual migraine), and aura (migraine with aura vs migraine without aura).

## Data management

**Data quality control.** Random data check will be conducted during data extraction and data entry to alert on erroneous, missing or out-of-range values and logical inconsistencies. Prompt verification and remedial action will be carried out, where necessary.

**Data storage and process.** The original CRFs will be stored in a locked cabinet for at least five years after the completion of the study. All data will be entered into a Microsoft Excel dataset. Electronic dataset will be stored in a password-protected computer drive. Only the research team members and the ethics committee members will have access to the data.

**Analytical methods.**    Data analysis will be conducted in relation to the study objectives:

1. To summarise participants' demographic characteristics, preferences and concerns, descriptive analyses will be conducted. Such descriptive analyses will be conducted in sub-cohorts of patients with various levels of predefined exposures, and compared analysis will be conducted, if appropriate. Categorical variables, including gender and satisfaction degree, will be presented using frequencies and percentages; quantitative variables such as disease duration and migraine frequency will be described with mean values and standard deviation.

2. To display the utilisation pattern of CHM for migraine and to summarise patient-reported MBS, cluster analyses and machine learning algorithms such as Apriori Algorithm [50] will be performed. Apriori Algorithm is a common method to identify core herb pairs and herb combinations [51–53]. Multiple correlation analysis and multivariable regression analysis will be conducted to explore factors related to clinical outcomes, where possible.

3. To evaluate the effects of CHM for adult migraine, generalised estimated equation will be adopted to compare longitudinal differences of clinical outcomes. Disease risk score and instrumental variables will be introduced to control measured and unmeasured confounders, respectively. Missing data will be handled using R-MICE package for the quantitative outcome measures at each evaluation timepoint.

## Ethics considerations

Participants may experience minimal discomfort and inconvenience during the observation. The researcher will communicate with participants professionally to minimise any discomfort associated with their participation in the study. The researcher will also provide detailed answers to any questions raised by the participants related to their migraine and treatments throughout the study. No extra financial, physical nor social demand are anticipated in this study; no payments will be provided to participants. Risk of data leaks is prevented by the well-established data storage and management plan as approved by the Ethics Committee.

Though participants will not receive financial benefits for their participation in this study, they will get a comprehensive assessment of migraine and comorbidities, as well as a monthly summary of and report on their migraine condition. It is expected that the findings of this study will provide pragmatic evidence for future clinical practice for migraine in a real-world setting.

## Discussion

Evidence-based practice has been increasingly used in healthcare [54]. It expects shared clinical decision making that integrates the best available, current, valid and relevant evidence [55]. This real-world observational study will take into consideration of the three vital elements [56] of evidence-based practice [55], namely, CHM utilisation patterns derived from clinical experts' experience, patients' preferences and concerns, and current clinical evidence. Other relevant findings such as patients' quality of life, medication consumptions, sleep disturbances, will also be documented in this study. Hence, this study will provide comprehensive evidence of CHM for migraine in the context of evidence-based practice.

Although RCT evidence is the gold standard to assess the efficacy and safety of CHM, RCT data commonly has poor external validity, thus with limited generalisability in real-world practice. In addition to current RCT findings, this study will provide an extension of evidence from registry studies in clinical practice [36]. To fully capture the benefits of individualised

interventions, patient-reported outcomes are recommended to be included along with standardised outcome measures [36]. Compared with RCT evidence, this real-world study may involve a broad range of participants, including chronic migraine with medication overuse, and migraine with psychiatric comorbidities. Findings from this study will fill in data gaps of the effects of CHM for these subtypes of migraine patients. In addition, this observational study innovatively applies patient-reported MBS in addition to common migraine outcomes to provide more comprehensive clinical assessments.

Previously, Chinese medicine practitioners' experience of using CHM for migraine were reported in retrospective studies based on clinical records and published data [57–61]. It should be noted that, retrospective studies often introduce bias or confounding factors which may impact on the interpretation of their results [62]. Unlike previous studies, this prospective study will provide more reliable and relevant data based on structured outcome measures and regular data collection throughout the observational period.

Common limitations of registry-based cohort studies and real-world studies are inevitable and cannot be completely excluded in this research. These include the lack of run-in period and randomisation, the subsequent recall bias [63] and various confounders [64]. Disease risk score and instrumental variables could be applied to deal with the confounders, and caution should be paid during the interpretation of these findings. Finally, this study is designed as a single-centre cohort, and the geographic limitation should be noted.

## Supporting information

**S1 Checklist. SPIRIT 2013 checklist: Recommended items to address in a clinical trial protocol and related documents.**
(DOCX)

**S1 File. Case report forms.**
(DOCX)

**S2 File. Migraine diary.**
(DOCX)

**S3 File.**
(DOCX)

**S4 File.**
(DOCX)

## Acknowledgments

The authors would like to express sincere thanks to the research assistant (Zhenhui Mao), all participants and healthcare professionals involved in the study.

## Author Contributions

**Conceptualization:** Shaohua Lyu, Claire Shuiqing Zhang, Anthony Lin Zhang, Jingbo Sun, Charlie Changli Xue, Xinfeng Guo.

**Data curation:** Shaohua Lyu.

**Funding acquisition:** Jingbo Sun.

**Methodology:** Shaohua Lyu, Claire Shuiqing Zhang, Anthony Lin Zhang, Xinfeng Guo.

**Writing – original draft:** Shaohua Lyu, Claire Shuiqing Zhang, Anthony Lin Zhang, Jingbo Sun.

**Writing – review & editing:** Shaohua Lyu, Claire Shuiqing Zhang, Anthony Lin Zhang, Charlie Changli Xue, Xinfeng Guo.

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
