## [Decision Letter · Decision Letter 0]

5 Oct 2021

PONE-D-21-25486Migraine Patients Visiting Chinese Medicine Hospital: Protocol for a Prospective, Registry-Based, Real-World Observational Cohort StudyPLOS ONE

Dear Dr. Guo,

Thank you for submitting your manuscript to PLOS ONE. After careful consideration, we feel that it has merit but does not fully meet PLOS ONE’s publication criteria as it currently stands. Therefore, we invite you to submit a revised version of the manuscript that addresses the points raised during the review process.

Please carefully address all points raised by the reviewers.

We look forward to receiving your revised manuscript.

Kind regards,

Sarah Michiels

Academic Editor

PLOS ONE

Journal Requirements:

Reviewers' comments:

Reviewer's Responses to Questions

**Comments to the Author**

1. Does the manuscript provide a valid rationale for the proposed study, with clearly identified and justified research questions?

Reviewer #1: Partly

Reviewer #2: Yes

2. Is the protocol technically sound and planned in a manner that will lead to a meaningful outcome and allow testing the stated hypotheses?

Reviewer #1: Partly

Reviewer #2: Yes

3. Is the methodology feasible and described in sufficient detail to allow the work to be replicable?

Reviewer #1: No

Reviewer #2: Yes

4. Have the authors described where all data underlying the findings will be made available when the study is complete?

Reviewer #1: No

Reviewer #2: No

5. Is the manuscript presented in an intelligible fashion and written in standard English?

Reviewer #1: Yes

Reviewer #2: Yes

6. Review Comments to the Author

You may also provide optional suggestions and comments to authors that they might find helpful in planning their study.

Reviewer #1: Thanks for giving me the opportunity to review the manuscript titled "Migraine Patients Visiting Chinese Medicine Hospital: Protocol for a Prospective, Registry-Based, Real-World Observational Cohort Study". Migraine has indeed been acknowledged as one of the most burdensome disorders, and treatment and prevention should be considered as an important health priority. Although I acknowledge some of the the limitations of RCT in the evaluation of treatment effectiveness in some cases, these design remain the only research design, in which causal relations between the randomized exposure (i.e. treatment) and outcome (e.g. quality of life) can be evaluated and established. The authors state that "the research evidence obtained from previous explanatory RCTs and systematic reviews need to be updated with more observational real-world evidence" to accommodate some issues that arise when conducting an RCT, such as "Such design may not reflect the complexity of migraine and its comorbidities (34), nor reflect the real-world practice of CHM which is commonly tailored according to individual’s syndrome differentiation (35)".

The step from RCT to observational trials to address all issues within an RCT-design is rather drastic, and the existence of pragmatic trials is ignored. Might I suggest the authors to consider the following publication on the use of pragmatic trials:

Ford, I., & Norrie, J. (2016). Pragmatic trials. New England journal of medicine, 375(5), 454-463.

Notwithstanding the limitations of RCT, observational studies have a tendency to yield highly biased estimates. Inferring on the effectiveness of CHM (as stated in the objectives of the authors "evaluate the effectiveness and safety of individualized CHM for adult migraine in a real-world setting;" will be a tough - if not impossible - challenge based on the method that was presented in the protocol of the authors. Registries can indeed uncover patterns and help in understanding health care seeking behaviour of certain groups. They can also help in understanding and exploring potential trends, but lack the ability to infer on causal relations. Frameworks around causal inference in observational studies have been suggested (see Causal Inference, What If by Robins and Hernan), but require a solid theoretical hypothesis. I would suggest the authors to consider creating a DAG, in which they represent the theoretical background on how there study might infer on a causal relation between exposure-outcome.

In the data-analyses, the authors state "Multiple imputation will be carried out to deal with missing data". However,

there is no information on the methodology that will be used to reach this goal. Which packages will be used?

Why do you rely on the Missing at Random (MAR) hypothesis? What if the data is rather NMAR? The section also lacks references. The authors refer to the "Apriori Algorithm", but no reference is added with a description of this algorithm.

In summary: Although registries can provide important insights, they limit the ability to infer on effectiveness.

I would suggest the authors to consider a pragmatic trial or - in case a pragmatic trial is infeasible - I would suggest the authors to focus more on patterns in healthcare behaviour.

Reviewer #2: Dear authors,

It is necessary to describe in detail the nature of CHM that will be used in this research, to connect it with previous research on animal models or in the laboratory research. It is necessary to show the chemical composition of the drug, the chemical formula of the drug, the site of action of the drug, the mechanism of action of the drug, the potential interactions of CHM with drugs for various comorbidities.

Introduction

I ask the authors to remove the part of the sentence which says that according to the latest classification, the division of migraine in relation to menstruation was made (this statement is not contained in the current classification).

Explain what is meant by a poor prognosis in patients with comorbidities.

The sentence "Currently there is no cure for migraine" is unnecessary.

List all groups of drugs, and western groups of drugs and western treatments for migraine. It is not enough to list only NSAIDs, triptans, CCB, beta blockers.

Are these drugs also used for comorbidity or just for migraine. If yes, for which comorbidities and provide references that confirm these claims.

OBJECTIVES

STUDY DESIGN

The dose of the medicine that the patients will take be determined on the basis of what references? What previous studies have confirmed the effectiveness of that dose and not some other dose of the drug? On the basis of which studies the administrative methods, courses, forms of CHM will be determined and which previous studies have confirmed that it is the best method, course, form?

Is it correct to collect data for patients who were not in the study during the entire planned period?

INTERVENTIONS

It is necessary to describe in detail the nature of CHM that will be used in this research, to connect it with previous research on animal models or in the laboratory. It is necessary to state the chemical composition of the drug, the chemical formula of the drug, the site of action of the drug, the mechanism of action of the drug, the potential interactions of CHM with drugs for various comorbidities. Are these medications used for an acute migraine attack or to prevent an attack or both?

DATA COLLECTION

TREATMENTS, EXAMINATIONS AND DIAGNOSES:

Chinese medicine syndrome diagnosis - I would ask the authors to explain the meaning of this term in more detail.

SAMPLING METHOD AND SAMPLE SIZE

The number 400 is based on the number of views in which time period? Explain in more detail the selection of this number of patients.

7. PLOS authors have the option to publish the peer review history of their article (what does this mean?). If published, this will include your full peer review and any attached files.

Reviewer #1: **Yes: **Robby De Pauw

Reviewer #2: No

---

## [Author Response · Author response to Decision Letter 0]

28 Oct 2021

Comments from the Editor:

Response: The manuscript was resubmitted in compliance with PLOS ONE’s style requirements. 

2. We note that the grant information you provided in the “Funding Information” and “Financial Disclosure” sections do not match. 

Response: Thanks for pointing this out. We have checked the funding information and made them consistent. 

Response: The ethics statement written in sections other than the Methods (Line 113-114) has been deleted. 

Comments from Reviewer #1

1. The step from RCT to observational trials to address all issues within an RCT-design is rather drastic, and the existence of pragmatic trials is ignored. I would suggest the authors to consider a pragmatic trial or - in case a pragmatic trial is infeasible - I would suggest the authors to focus more on patterns in healthcare behaviour.

2. Inferring on the effectiveness of CHM (as stated in the objectives of the authors "evaluate the effectiveness and safety of individualized CHM for adult migraine in a real-world setting;" will be a tough - if not impossible - challenge based on the method that was presented in the protocol of the authors. I would suggest the authors to consider creating a DAG, in which they represent the theoretical background on how the study might infer on a causal relation between exposure-outcome.

Response to comments 1&2: 

Thanks for your comments and suggestions. 

We revised the objectives of this study (see Line 94-99). 

This study is an observational study, without any intervention being provided by the researcher. The purpose of this study is to explore the effects and safety of CHM treatments in the real-world clinical practice. Therefore, there is no causal inference would be made based on our study results.

However, the findings from this observational study will be invaluable for the future design of a pragmatic trial. 

3. In the data-analyses, the authors state "Multiple imputation will be carried out to deal with missing data". However, there is no information on the methodology that will be used to reach this goal. Which packages will be used?

Why do you rely on the Missing at Random (MAR) hypothesis? What if the data is rather MNAR? 

Response:

In this observational study, we would like to present the overall findings in real-world clinical setting as much as possible. Missing data will be handled using R-MICE package for the following outcome measures (migraine frequency, average pain intensity NRS, MSQ, GAD-7, PHQ-9, ISI) at each evaluation timepoint within the 12 weeks’ observation period. 

As mentioned above, this observational study aims to explore the effects of CHM for migraine, summarise patterns of CHM for migraine and the characteristics of migraine patients, we will present what we observed for the patterns of CHM and the characteristics of patients.

The “Analytical methods” section has been revised to clarify this information (Line 244-262). 

4. The section also lacks references. The authors refer to the "Apriori Algorithm", but no reference is added with a description of this algorithm

Response: 

The reference and brief description have been added as suggested (Line 254-256).

Comments from reviewer #2

Introduction

1. I ask the authors to remove the part of the sentence which says that according to the latest classification, the division of migraine in relation to menstruation was made (this statement is not contained in the current classification).

Response: 

Thanks for the suggestion. This statement has been removed. 

2. Explain what is meant by a poor prognosis in patients with comorbidities.

Response:

An explanation about the “poor prognosis” was added (see Line 57-58).

3. The sentence "Currently there is no cure for migraine" is unnecessary.

List all groups of drugs, and western groups of drugs and western treatments for migraine. It is not enough to list only NSAIDs, triptans, CCB, beta blockers.

Are these drugs also used for comorbidity or just for migraine. If yes, for which comorbidities and provide references that confirm these claims.

Response: 

The sentence of "Currently there is no cure for migraine" was deleted.

All groups of western drugs for migraine and the devices are listed in Line 62-71.

Specific drugs which can be adopted for migraine comorbidities as recommended by clinical guidelines were also listed in this paragraph. 

STUDY DESIGN

4. The dose of the medicine that the patients will take be determined on the basis of what references? What previous studies have confirmed the effectiveness of that dose and not some other dose of the drug? On the basis of which studies the administrative methods, courses, forms of CHM will be determined and which previous studies have confirmed that it is the best method, course, form?

Response: 

This is an observational study, the treatments for the patients in this study are prescribed by the clinicians according to each patient’ real situation. These interventions are not predefined nor interfered by the observation. We are recording and summarising what we observe in the real-world clinical practice. 

5. Is it correct to collect data for patients who were not in the study during the entire planned period?

Response: 

This study only collect data from patients who meet the inclusion and exclusion criteria. 

INTERVENTIONS

6. It is necessary to describe in detail the nature of CHM that will be used in this research, to connect it with previous research on animal models or in the laboratory. It is necessary to state the chemical composition of the drug, the chemical formula of the drug, the site of action of the drug, the mechanism of action of the drug, the potential interactions of CHM with drugs for various comorbidities. 

Response: 

As mentioned above, in this observational study, the CHM interventions are tailored based on each patient’ symptoms and need, and decided by the clinicians, who are independent from the researchers. The CHM treatments to be observed are not able to be anticipated, therefore we are not able to provide the required information in this protocol, but this will be addressed when we report the results of this study. Upon the completion of this observational study, we will identify the most frequent herbs, the core herb pairs and herb combinations being used in clinical practice, and then provide information on their mechanisms for migraine in the final study report. 

7. Are these medications used for an acute migraine attack or to prevent an attack or both?

Response: 

The western medications and CHM formulations being prescribed by clinicians can be used for either acute migraine attacks or migraine prevention. 

DATA COLLECTION

TREATMENTS, EXAMINATIONS AND DIAGNOSES:

8. Chinese medicine syndrome diagnosis - I would ask the authors to explain the meaning of this term in more detail.

Response: 

Chinese medicine syndrome diagnosis is the process of a comprehensive analysis of patients’ symptom and signs, under the principles of Chinese medicine aetiology and pathogenesis. It is used to guide the selection of Chinese medicine treatments. 

Based on the Chinese medicine syndrome diagnosis, treatments will be individualised because patients may present different syndromes although they are diagnosed with the same clinical condition; and each patient may present different syndromes in different stages of a whole disease course, therefore the treatment will also be modified accordingly. 

SAMPLING METHOD AND SAMPLE SIZE

9. The number 400 is based on the number of views in which time period? Explain in more detail the selection of this number of patients.

Response: 

The estimated number of participants is approximately 400 based on the average annually outpatient visit numbers at the headache clinic of GPHCM between July 2018 and July 2020.

This information has been clarified in the Methods section (see Line 221-222).

End.

---

## [Decision Letter · Decision Letter 1]

24 Feb 2022

Migraine Patients Visiting Chinese Medicine Hospital: Protocol for a Prospective, Registry-Based, Real-World Observational Cohort Study

PONE-D-21-25486R1

Dear Dr. Guo,

We’re pleased to inform you that your manuscript has been judged scientifically suitable for publication and will be formally accepted for publication once it meets all outstanding technical requirements.

Kind regards,

Sarah Michiels

Academic Editor

PLOS ONE

Journal Requirements:

1. Your ethics statement should only appear in the Methods section of your manuscript. If your ethics statement is written in any section besides the Methods, please delete it from any other section. 

Reviewers' comments:

Reviewer's Responses to Questions

**Comments to the Author**

1. Does the manuscript provide a valid rationale for the proposed study, with clearly identified and justified research questions?

Reviewer #2: Partly

2. Is the protocol technically sound and planned in a manner that will lead to a meaningful outcome and allow testing the stated hypotheses?

Reviewer #2: Yes

3. Is the methodology feasible and described in sufficient detail to allow the work to be replicable?

Reviewer #2: Yes

4. Have the authors described where all data underlying the findings will be made available when the study is complete?

Reviewer #2: Yes

5. Is the manuscript presented in an intelligible fashion and written in standard English?

Reviewer #2: Yes

6. Review Comments to the Author

You may also provide optional suggestions and comments to authors that they might find helpful in planning their study.

Reviewer #2: Thank you for the answers. Keep in mind my comments when conducting research. I suggest that the paper should be accepted.

7. PLOS authors have the option to publish the peer review history of their article (what does this mean?). If published, this will include your full peer review and any attached files.

Reviewer #2: No

---

## [Editor Report · Acceptance letter]

7 Mar 2022

PONE-D-21-25486R1 

Migraine Patients Visiting Chinese Medicine Hospital: Protocol for a Prospective, Registry-Based, Real-World Observational Cohort Study 

Dear Dr. Guo:

I'm pleased to inform you that your manuscript has been deemed suitable for publication in PLOS ONE. Congratulations! Your manuscript is now with our production department. 

Kind regards, 

on behalf of

Prof. Sarah Michiels 

Academic Editor

PLOS ONE